

# Validation of endogenous reference genes in rat cerebral cortex for RT-qPCR analyses in developmental toxicity studies

Louise Ramhøj, Marta Axelstad and Terje Svingen

Division of Diet, Disease Prevention and Toxicology, National Food Institute, Technical University of Denmark, Kongens Lyngby, Denmark

## ABSTRACT

Relative gene expression data obtained from quantitative RT-PCR (RT-qPCR) experiments are dependent on appropriate normalization to represent true values. It is common to use constitutively expressed endogenous reference genes (RGs) for normalization, but for this strategy to be valid the RGs must be stably expressed across all the tested samples. Here, we have tested 10 common RGs for their expression stability in cerebral cortex from young rats after in utero exposure to thyroid hormone (TH) disrupting compounds. We found that all 10 RGs were stable according to the three algorithms geNorm, NormFinder and BestKeeper. The downstream target gene *Pvalb* was significantly downregulated in brains from young rats after in utero exposure to propylthiouracil (PTU), a medicinal drug inhibiting TH synthesis. Similar results were obtained regardless of which of the 10 RGs was used for normalization. Another potential gene affected by developmental TH disruption, *Dio2*, was either not affected, or significantly upregulated about 1.4-fold, depending on which RG was used for normalization. This highlights the importance of carefully selecting correct RGs for normalization and to take into account the sensitivity of the RT-qPCR method when reporting on changes to gene expression that are less than 1.5-fold. For future studies examining relative gene expression in rat cerebral cortex under toxicological conditions, we recommend using a combination of either *Rps18/Rpl13a* or *Rps18/Ubc* for normalization, but also continuously monitor any potential regulation of the RGs themselves following alterations to study protocols.

# INTRODUCTION

Exposure to thyroid hormone (TH) disruptors during pregnancy can have detrimental consequences for the child's brain development. In the developing brain, THs exert their action by regulating gene transcription required for normal development and function. If TH levels or action is disturbed, it can lead to dysregulated gene expression and ultimately cause structural and functional changes to the brain (*Bernal, 2017*; *Gilbert & Zoeller, 2011*). Worryingly, numerous compounds that humans are exposed to can also disturb TH levels or action. Understanding how these compounds can cause adverse health effects is thus of great importance and will help us conceive new testing strategies

Corresponding author
Terje Svingen, tesv@food.dtu.dk

designed to capture harmful compounds before they potentially are used in ways that may harm the developing child.

Various effect biomarkers are currently being explored in an effort to devise more sensitive and reliable testing strategies to detect adverse effects on brain developmental caused by developmental TH disruption. For instance, measurements of TH levels have recently been included in several OECD test guidelines used for toxicity testing of environmental chemicals, such as TG 421 (*OECD, 2016a*), TG 422 (*OECD, 2016b*) and TG 414 (*OECD, 2018*). Adverse outcome pathways (AOPs) are also being developed to improve our understanding of how different molecular initiating events can cause TH insufficiency in the brain and lead to adverse neurodevelopmental effects. One knowledge gap in these AOPs is the specific relationship between TH-dependent gene expression and subsequent effects on the developing nervous system (*Crofton et al., 2017*). Thus, future studies should aim to characterize this relationship at the molecular level.

Despite great advances in different transcriptomics approaches (*Darde, Chalmel & Svingen, 2018*), quantitative RT-PCR (RT-qPCR) remains a popular method for determining relative gene expression levels, especially when only a few target genes are examined. RT-qPCR is considered a reliable and cheap way of quickly determining relative transcript abundance without having to perform more complex bioinformatics post-run analyses. Most often, RT-qPCR results are simply converted into relative fold expression data by the comparative Ct method (*Schmittgen & Livak, 2008*). Although theoretically a sound method, it does rely on certain assumptions that must be adhered to if the data are to be true representations of the transcriptional landscape of the cells or tissues. These assumptions include stable RNA integrity across samples, correct primer design, amplification efficiencies close to 100% unless absolute quantification methodologies are implemented, and correct normalization of input RNA levels (*Nolan, Hands & Bustin, 2006*). These are all critical parameters, but normalization continues to be an overlooked issue, especially when using endogenous reference genes (RGs) for normalization (*Piller, Decosterd & Suter, 2013*).

Before exploring the utility of using changes to gene expression levels in the rodent brain as an adverse effect readout for TH disruption, it is imperative that we first identify RGs that are relatively unchanged by exposure to putative TH disrupting chemicals, not least if relative quantification methods such as RT-qPCR are to be used. Although there are studies that have examined stability of common RGs in rat brains at various developmental stages or following insult (*Al-Bader & Al-Sarraf, 2005*; *Cook et al., 2009*; *Julian et al., 2014*; *Swijsen et al., 2012*; *Zhou et al., 2016*), a systematic evaluation of suitable RGs in rat brains following developmental TH disruption is lacking. In this study, we have tested the stability of 10 common RGs in the cerebral cortex of rat offspring after exposure to the TH synthesis inhibitor drug propylthiouracil (PTU), as well as two putative TH disruptors: the UV filter octyl methoxycinnamate (OMC) and the flame retardant pentabromodiphenyl ether mixture, DE-71.

# MATERIALS AND METHODS
## Chemicals
Chemicals used in this study were: PTU (propylthiouracil or 6-propyl-2-thiouracil), Cas no 51-52-5, product number P3755, purity >99% (Sigma-Aldrich, Brøndby, Denmark);

DE-71 (penta-BDE, lot 7550OK20A), generous donation from Dr. Kevin Crofton (U.S. Environmental Protection Agency, Washington, D.C., USA); OMC (2 ethylhexyl 3-(4-methoxyphenyl)-2 propenoate or 2-ethylhexyl-4-methoxycinnamate), CSA no 5466-77-3, product number ACR291160250, purity 98.0% (VWR Bie & Bernstsen, Herlev, Denmark); corn oil as control compound and vehicle (Sigma-Aldrich, Brøndby, Denmark).

## Animals

The animal study was approved by the Danish Animal Experiments Inspectorate (authorization No. 2015-15-0201-00553 C3) and overseen by the National Food Institute's in-house Animal Welfare Committee for animal care and use. Time-mated Sprague-Dawley rats (Charles River, Germany) were gavaged once daily with either vehicle (corn oil), 2.5 mg/kg bw/day PTU, 40 mg/kg bw/day DE-71 or 500 mg/kg bw/day OMC from gestational day 7 through to postnatal day 22. This study includes seven animals from each group; a total of 28 animals from 28 different litters. On PD 16 offspring were decapitated under $CO_2/O_2$-anesthesia. Brains were collected from one male offspring per litter and an oblique slab of anterior to lateral cortex of each hemisphere collected in RNAlater and stored at $-80\ °C$ until RNA extraction.

## RNA extraction, cDNA synthesis and quantitative RT-PCR

About 25–40 mg tissue from rat cerebral cortex was homogenized using a Tissue Lyser II (Qiagen, Hilden, Germany), then total RNA extracted using an RNeasy Mini kit (Qiagen, Hilden, Germany) including on-column DNaseI treatment, as per manufacturer's instruction. RNA quantity and quality were analyzed on a nano-drop spectrophotometer as well as with the Agilent 2100 Bioanalyzer system and Eukaryote Total RNA Nano assay (Agilent Technologies, Santa Clara, CA, USA). All samples had a RIN score >8. cDNA was synthesized from 500 ng RNA using a Omniscript kit (Qiagen, Hilden, Germany) and random primer mix (New England Biolabs, Ipswich, MA, USA) according to manufacturer's instructions. RT-qPCR experiments were run in technical duplicates on a Quantstudio 7 Flex Real-Time PCR system (Applied Biosystems, Foster City, CA, USA; Thermo Fischer Scientific, Waltham, MA, USA) on 384-well plates with 11 µl reactions containing TaqMan Fast Universal Mastermix (Life Technologies, Carlsbad, CA, USA), TaqMan Gene Expression Assays as listed in Table 1 (Life Technologies, Carlsbad, CA, USA), and three µl cDNA diluted 1:20 from stock. The variation between technical duplicates did not exceed 0.5 cycles. PCR cycling conditions were 95 °C for 20 s, followed by 45 two-step thermal cycles of 95 °C for 1 s and 60 °C for 20 s. Data were acquired with the Quantistudio 7 Flex software and are reported in Dataset S1.

## Analytical methods and statistics

Three different algorithms were used to analyze expression stability of the 10 RGs. The first, geNorm (*Vandesompele et al., 2002*), calculates an expression stability value (*M*) of a single gene based on the average pairwise variation between all of the RGs included. Stepwise exclusion of the least stable gene result in the selection of the two most stable genes. For homogenous samples, an *M*-value below 0.5 is considered "stable," whereas an

**Table 1 List of TaqMan assays.**

| Gene | RefSeq | Name | TaqMan |
|------|--------|------|--------|
| Reference genes | | | |
| *Actb* | NM_031144 | Beta-actin | Rn00667869 |
| *B2m* | NM_012512 | Beta-2 microglobulin | Rn00560865 |
| *Gapdh* | NM_017008 | Glyceraldehyde-3-phosphate dehydrogenase | Rn01775763 |
| *Hprt1* | NM_012583 | Hypoxanthine guanine phosphoribosyl transferase | Rn01527840 |
| *Rpl13a* | NM_173340 | Ribosomal protein L13A | Rn00821946 |
| *Rps18* | NM_213557 | Ribosomal protein S18 | Rn01428913 |
| *Rps29* | NM_012876 | Ribosomal protein S29 | Rn00820645 |
| *Sdha* | NM_130428 | Succinate dehydrogenase complex, subunit A, flavoprotein (Fp) | Rn00590475 |
| *Tbp* | NM_001004198 | TATA box binding protein | Rn01455646 |
| *Ubc* | NM_017314 | Ubiquitin C | Rn01789812 |
| Additional genes | | | |
| *Dio2* | NM_031720 | Deiodinase, iodothyronine, type II | Rn00581867 |
| *Pvalb* | NM_022499 | Parvalbumin | Rn00574541 |

$M$-value of 1.0 is considered "stable" for heterogenous samples. The second, NormFinder (*Andersen, Jensen & Ørntoft, 2004*), uses transformed Ct data to estimate an expression stability score ($S$) by combining intra- and inter-group variations for each of the RGs. The NormFinder algorithm accounts for biological heterogeneity and co-regulation of genes. An $S$ score below 0.5 is considered "stable." The third, BestKeeper (*Pfaffl et al., 2004*), performs parametric tests on normally distributed expression levels of each RG using untransformed Ct values. It estimates the geometric means of the Ct values, determines coefficient of variance (CV) and the Pearson's correlation ($r$) for each RG, and uses standard deviation (SD) values to create a weighted index of most stable RG across the samples. The lower the SD and CV, the more stable the RG is expressed across the biological samples with a suitable RG having an SD <1.0.

Relative gene expression values were obtained by using the comparative Ct method (Applied Biosystems Research Bulletin No. 2 P/N 4303859) (*Schmittgen & Livak, 2008*). Statistical analysis of RT-qPCR data was performed in Graph Pad Prism v8 (GraphPad Software, San Diego, CA, USA) using two-tailed, unpaired $t$-test. Statistical significance is denoted $^*p < 0.05$, $^{**}p < 0.01$, $^{***}p < 0.001$, $^{****}p < 0.0001$.

## RESULTS

In RT-qPCR experiments, it is advisable to first normalize input RNA levels by using the same amount of RNA for cDNA synthesis across all samples to be included in the same analysis. We used 500 ng of total RNA for each sample and synthesized cDNA from the same master mix. Following RT-qPCR amplification, we acquired the raw mean Ct values from technical duplicate reactions and did a first-round assessment of relative mRNA abundance and stability across samples for 10 putative endogenous RGs. Looking across all the 28 samples representing four different exposure groups, the SD did not

**Table 2 Mean RT-qPCR threshold (Ct) values of 10 RGs in juvenile (16 days postnatal) rat cerebral cortex from four in utero exposure groups.**

| Gene | "Vehicle" control (Mean ± SD) | PTU 2.5 mg/kg bw/d (Mean ± SD) | OMC 500 mg/kg bw/d (Mean ± SD) | DE-71 40 mg/kg bw/d (Mean ± SD) | All (Mean ± SD) |
|------|-------------------------------|--------------------------------|--------------------------------|---------------------------------|------------------|
| B2m | 22.69 ± 0.30 | 22.72 ± 0.29 | 22.71 ± 0.42 | 22.76 ± 0.26 | 22.72 ± 0.32 |
| Actb | 20.39 ± 0.48 | 20.30 ± 0.50 | 20.03 ± 0.32 | 20.42 ± 0.20 | 20.29 ± 0.42 |
| Gapdh | 20.73 ± 0.45 | 20.90 ± 0.37 | 20.67 ± 0.28 | 20.80 ± 0.26 | 20.77 ± 0.36 |
| Hprt | 24.98 ± 0.47 | 24.95 ± 0.35 | 24.70 ± 0.22 | 24.91 ± 0.20 | 24.88 ± 0.35 |
| Rpl13a | 23.99 ± 0.26 | 24.13 ± 0.38 | 23.83 ± 0.24 | 24.01 ± 0.13 | 23.99 ± 0.29 |
| Rps18 | 21.63 ± 0.19 | 21.57 ± 0.25 | 21.36 ± 0.20 | 21.62 ± 0.18 | 21.54 ± 0.23 |
| Rps29 | 21.43 ± 0.23 | 21.34 ± 0.16 | 21.20 ± 0.22 | 21.45 ± 0.13 | 21.36 ± 0.21 |
| Sdha | 24.00 ± 0.32 | 24.28 ± 0.45 | 23.82 ± 0.19 | 24.02 ± 0.21 | 24.03 ± 0.35 |
| Tbp | 29.19 ± 0.34 | 29.39 ± 0.42 | 29.14 ± 0.13 | 29.27 ± 0.23 | 29.25 ± 0.31 |
| Ubc | 21.34 ± 0.32 | 21.39 ± 0.24 | 21.36 ± 0.19 | 21.42 ± 0.12 | 21.38 ± 0.23 |

**Notes:**
Control, exposed to 2.5 mg/kg$_{bw}$ PTU, 500 mg/kg$_{bw}$ OMC, or 40 mg/kg$_{bw}$ DE-71. Mean ± SD was calculated from seven biological replicates ($N = 7$; All, $N = 28$). bw/d, body weight per day.

exceed 0.42 cycles, and with a maximum SD of 0.5 in the PTU group (Table 2). This indicates a high level of expression stability for all included RGs in the cortex of young rats.

We used three different algorithms to assess the relative expression stability of the 10 RGs: geNorm (*Vandesompele et al., 2002*), NormFinder (*Andersen, Jensen & Ørntoft, 2004*) and BestKeeper (*Pfaffl et al., 2004*). With some exceptions, the stability ranking was similar between the three algorithms, all scoring *Rps18*, *Rpl13a* and *Sdha* in the top five and *Gapdh*, *B2m* and *Actb* in the bottom five (Table 3). Notably, all 10 RGs were deemed suitable ("stable enough") for use as single endogenous normalizing genes when performing RT-qPCR analysis with the comparative Ct method on rat cortex tissue under the experimental conditions as described in this study. For relatively homogenous tissues, the cut-off *M*-value for geNorm is <0.5, whereas the worst performer tested herein, *Actb*, had an *M*-value of 0.023. Likewise, NormFinder recommends a cut-off *S*-value of <0.5 and the highest *S*-values obtained in our experiment was 0.081 for *B2m*. Finally, BestKeeper recommends that RGs should have a SD value <1.0 to be deemed suitable for normalization and herein the highest SD value was 0.32 for *Actb*.

It is common to use only one endogenous RG to normalize RT-qPCR data, although this has a higher risk of resulting in erroneous normalized expression data. Using two or more RGs for normalization minimizes this problem and is recommended, provided they are all stably expressed across the examined samples (*Vandesompele et al., 2002*). The geNorm and NormFinder algorithms also provide suggestions for best combinations of RGs, which not necessarily comprise those that are ranked at the very top of the list. Across the samples tested in this study, geNorm suggests using a combination of *Rps18* and *Rpl13a*, whereas NormFinder suggests combining *Rps18* and *Ubc* (Fig. 1). Notably, *Rps18* was ranked second best by all three algorithms as single-use RG and was included in both RG pairs.

**Table 3 Stability ranking of 10 RGs determined by GeNorm, NormFinder and BestKeeper.**

| Rank | geNorm | | NormFinder | | BestKeeper | | |
|---|---|---|---|---|---|---|---|
| | Gene | *M*-value | Gene | *S*-value | Gene | SD | CV |
| 1 | *Tbp* | 0.0150 | *Hprt* | 0.0467 | *Rps29* | 0.1788 | 0.8372 |
| 2 | *Rps18* | 0.0159 | *Rps18* | 0.0533 | *Rps18* | 0.1888 | 0.8762 |
| 3 | *Hprt* | 0.0155 | *Ubc* | 0.0574 | *Ubc* | 0.2002 | 0.9364 |
| 4 | *Sdha* | 0.0167 | *Sdha* | 0.0582 | *Rpl13a* | 0.2047 | 0.8533 |
| 5 | *Rpl13a* | 0.0170 | *Rpl13a* | 0.0594 | *Sdha* | 0.2358 | 0.9812 |
| 6 | *Ubc* | 0.0174 | *Tbp* | 0.0611 | *Tbp* | 0.2375 | 0.8120 |
| 7 | *Rps29* | 0.0189 | *Gapdh* | 0.0624 | *Hprt* | 0.2552 | 1.0255 |
| 8 | *Gapdh* | 0.0195 | *Rps29* | 0.0699 | *B2m* | 0.2726 | 1.2001 |
| 9 | *B2m* | 0.0220 | *Actb* | 0.0770 | *Gapdh* | 0.2868 | 1.3808 |
| 10 | *Actb* | 0.0231 | *B2m* | 0.0813 | *Actb* | 0.3234 | 1.5939 |

**Note:**

Ranking is depicted in descending order, so that 1 is most stable and 10 is least stable. geNorm calculates an average expression stability (*M*-value), with a cut-off value of <0.5 for relatively homogenous samples. NormFinder calculates relative expression stability (*S*-value), also with a cut-off of <0.5. BestKeeper calculates standard deviation (SD) and coefficient of variance (CV), with a SD cut-off of <1.0.

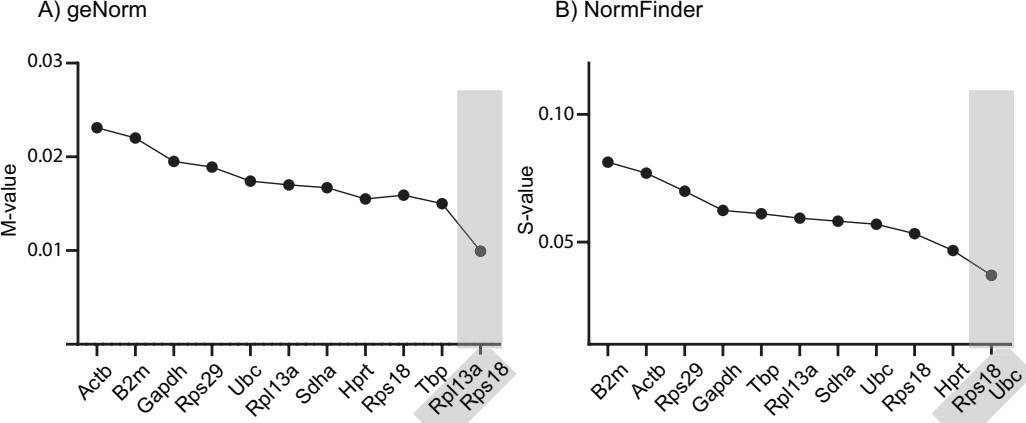

**Figure 1 Stability ranking of RGs according to (A) geNorm and (B) NormFinder, including best paired combination.** The 10 RGs were tested on rat cerebral cortex in control and in offspring after in utero exposure to TH-disrupting compounds PTU, OMC or DE-71. The higher the stability scores (*M*- or *S*-values), the less stable the gene is expressed across groups. Proposed cut-off values for both algorithms is 0.5, so all of the included RGs are deemed "suitable" for normalization purposes. The analyses also included the determination of the best pair of RGs recommended for normalization of RT-qPCR data (shaded). *N* = 7 per group.

To test the performance of the 10 RGs, we performed relative fold expression analysis on two genes, *Pvalb* and *Dio2*, as they could potentially be dysregulated after exposure to PTU and other chemicals. Both genes were normalized by the comparative Ct method using all of the 10 RGs individually (Fig. 2). *Pvalb* was significantly downregulated in the cortex of PTU-exposed offspring (around threefold) regardless of which of the 10 RGs that was used for normalization of input RNA. *Dio2* was unchanged in the exposed tissue relative to controls when performing the analysis with most of the RGs; however, it was significantly up-regulated (up to about 1.4-fold) when *Rpl13a*, *Sdha* or *Tbp* was used for

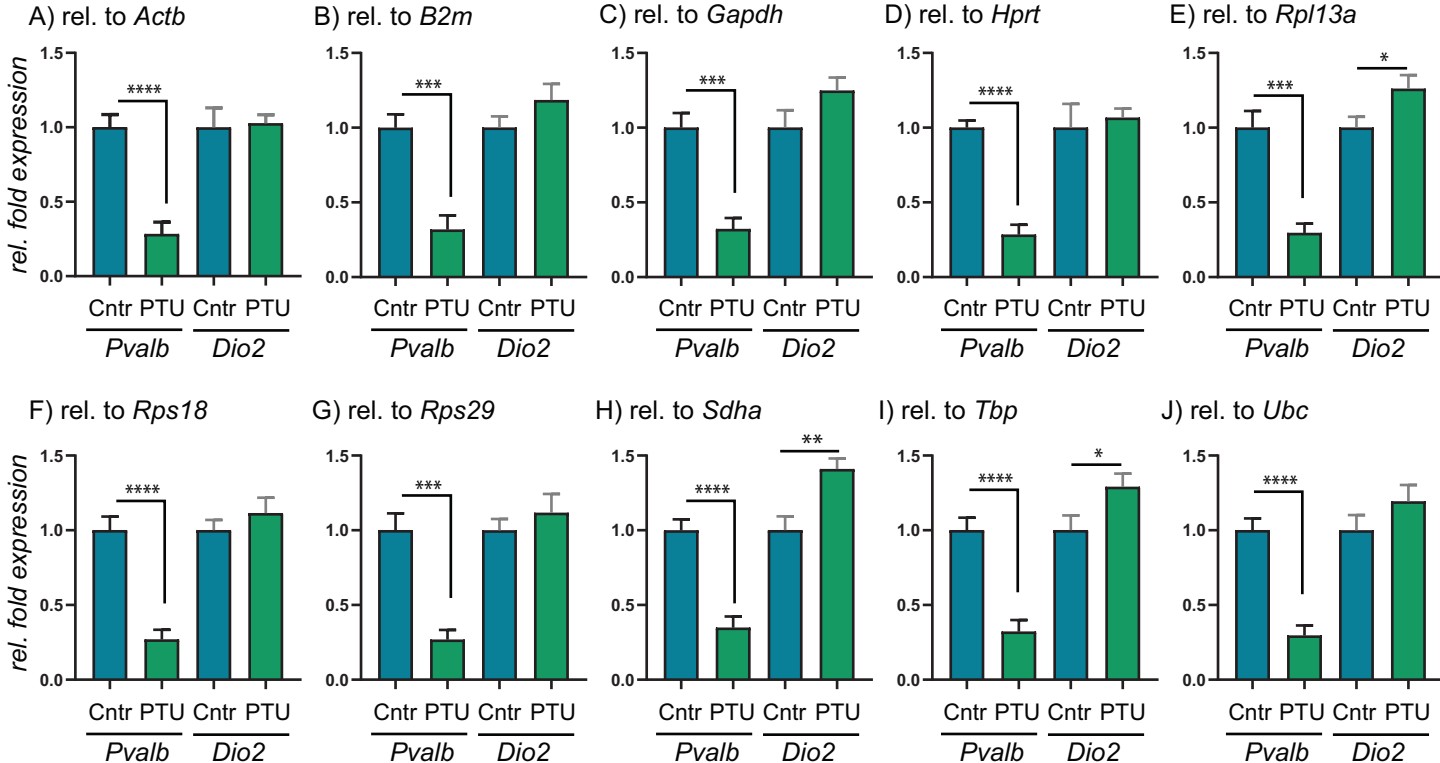

**Figure 2 Relative expression levels of *Pvalb* and *Dio2* in cerebral cortex of juvenile male rats after in utero exposure to PTU, normalized by different RGs.** (A–J) Cerebral cortex tissue from control and PTU-exposed offspring was analyzed for changes to relative mRNA abundance of two genes downstream of TH signaling, *Pvalb* and *Dio2*. The same RT-qPCR data were analyzed by normalizing against 10 different RGs that all were deemed suitable for normalization purposes by three algorithms, geNorm, NormFinder and BestKeeper. In all instances, *Pvalb* was found to be significantly downregulated in the PTU group. Only in three out of 10 instances was *Dio2* found to be statistically significantly upregulated in the PTU group, with the remaining seven RGs giving no changes to mRNA abundance. $N = 7$ per group; $^*p < 0.05$, $^{**}p < 0.01$, $^{***}p < 0.001$, $^{****}p < 0.00001$.

normalization. When normalizing data using the best pair combinations suggested by geNorm or NormFinder (Fig. 3), *Pvalb* was again significantly downregulated (about threefold), whereas *Dio2* was unchanged between control and PTU-exposed cortex tissue, albeit the mean values showed a trend toward being upregulated in the exposed group. Comparable expression patterns were observed between the two pairs of RGs for OMC and DE-71 exposed groups.

## DISCUSSION

Quantitative RT-PCR remains a popular method for analyzing gene expression levels in cells and tissues. It has developed into a fast, cheap and reliable method to quantify the abundance of RNA species. However, data analysis relies heavily on specific assumptions, not least the use of stable internal normalizers for the popular comparative Ct method (*Nolan, Hands & Bustin, 2006*). Since using endogenous RGs is the most common way of normalizing inter-sample variations, there has been an increasing focus on testing RG stability across different tissues under various experimental conditions; and this has proven to be extremely important, as many "historical housekeeping genes"

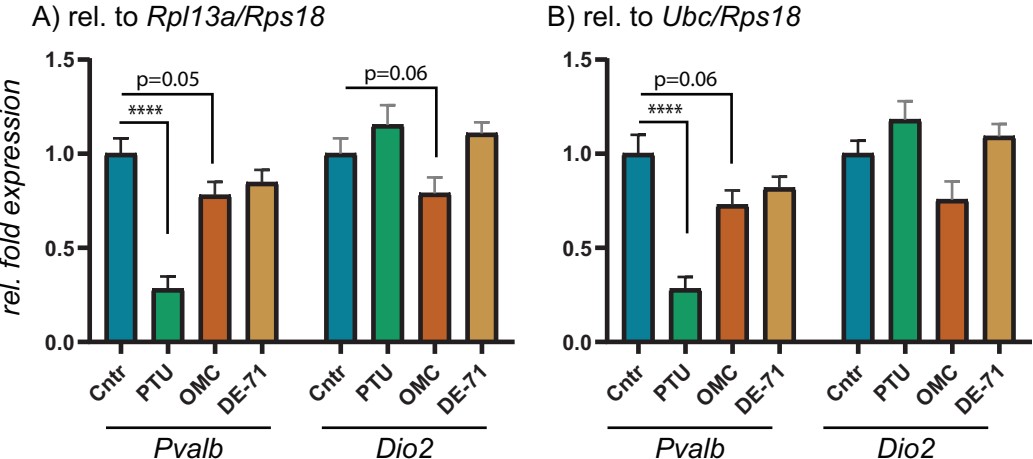

**Figure 3** **Relative expression levels of *Pvalb* and *Dio2* in cerebral cortex of juvenile male rats after in utero exposure to PTU, OMC, or DE-71 normalized by optimal RG pairs.** Cerebral cortex tissue from control and PTU-, OMC- and DE-71-exposed offspring was analyzed for changes to relative mRNA abundance of two genes downstream of TH signaling, *Pvalb* and *Dio2*. The same RT-qPCR data was analyzed by normalizing against the RG pair (A) *Rpl13/Rps18* or (B) *Ubc/Rps18*, which were suggested to be optimal by two algorithms, geNorm and NormFinder. $N = 7$ per group; ****$p < 0.00001$.

have in many instances been shown to be highly unstable (*Piller, Decosterd & Suter, 2013*).

In this study, we tested 10 common endogenous RGs for their stability in cerebral cortex of young male rats after in utero exposure to three TH-disrupting chemicals: PTU, OMC and DE-71. Expression stability was tested across the three exposure groups in addition to control animals. Following PCR cycling, the relative expression stability was remarkably good for all of the 10 RGs. According to three different algorithms designed to calculate relative expression stability—geNorm, NormFinder and BestKeeper—all of the 10 RGs obtained stability scores well inside recommended cut-off values, in essence predicting all of the RGs to be suitable for RT-qPCR normalization. This is in contrast to previous studies testing across different rat tissues (*Svingen et al., 2015*) or brain tissue from rats (*Cook et al., 2009*; *Yang et al., 2008*; *Zhou et al., 2016*), mice (*Crans et al., 2019*) or humans (*Rydbirk et al., 2016*). Notably, brain tissue from a rat sleep apnea model displayed high stability of all tested RGs except 18S rRNA (*Julian et al., 2014*), a stability also seen in rat brains across fetal developmental stages (*Al-Bader & Al-Sarraf, 2005*), which agrees with our findings of relative high stability among RGs in this tissue (*Julian et al., 2014*). There are, however, studies showing unstable expression of common RGs depending on experimental parameters and what brain regions are examined (*Cook et al., 2009*; *Swijsen et al., 2012*; *Yang et al., 2008*; *Zhou et al., 2016*).

geNorm suggests that the stability value $M$ should be lower than 0.5 for homogenous tissues in order to be used as endogenous normalizing gene. In our study, *Actb* was the worst performer yet obtained an $M$-value of 0.023, which is suggestive of high stability between groups. Similar scores were obtained with the other two algorithms. Based on these scores, it can be concluded that all the 10 RGs tested herein are suitable for

normalization of rat cortex under the described experimental conditions. However, we would still recommend using at least two RGs when performing RT-qPCR analyses, also in accordance with previous recommendations (*Vandesompele et al., 2002*). This point is particularly apparent in view of the data obtained for the proof-of-principle gene expression profiles that we carried out for *Pvalb* and *Dio2*.

To scrutinize the relative expression data obtained by using the comparative Ct method depending on which endogenous RG that was used, we performed RT-qPCR experiments on the same brain tissues for *Pvalb* and *Dio2*. These two genes were chosen based on previous reports showing dysregulation in rat brain following exposure to TH disrupting chemicals. For instance, to compensate for low brain T3 *Dio2* expression levels may increase for increased T4 deiodination to maintain T3 (*Morse et al., 1993*; *Peeters et al., 2001*; *Sharlin et al., 2010*), whereas *Pvalb* may be a suitable biomarker for TH disruption in the brain; *Pvalb* expression has been correlated to serum T4 levels (*O'Shaughnessy et al., 2018*). Correspondingly, we found that *Pvalb* was significantly downregulated to about threefold in the PTU-exposed rats compared to controls. This result is unequivocal irrespective of which of the 10 RGs was used for normalization, albeit with small variations in relative means and *p*-values. *Dio2*, on the other hand, was significantly upregulated when normalizing data with either *Rpl13a*, *Sdha* or *Tbp*, whereas the tending upregulation was not statistically significant if the data were normalized with any of the remaining seven RGs; nor with the recommended pairs of RGs: *Rpl13a*/*Rps18* or *Rps18*/*Ubc*. This can seem counter-intuitive since all of the RGs should be stably expressed across the samples in question, but the answer may lie more with the uncertainty of determining significant changes in gene expression if they are of a smaller magnitude.

When performing RT-qPCR experiments, they are typically run in duplicate or triplicate reactions, so-called technical replicates. It is common to allow for variation of 0.5–1.0 Ct within technical replicates, which in itself will impact on the relative sensitivity of the assay. When transforming data, as is done with the comparative Ct method, a difference of 0.5 Ct (comparable to delta-Ct between samples) will equal about 1.4-fold difference in relative expression. Thus, it can become problematic to reliably determine changes in relative expression less than 50% up or down regardless of statistical *p*-value. This sensitivity parameter should be considered when reporting data with small changes to relative fold expression, as for instance is the case with the *Dio2* data presented in Fig. 2. Notably, *Dio2* was not statistically significantly altered relative to control when normalizing the data with the recommended pairs of RGs.

For future studies looking at the effects on gene expression in cerebral cortex of rat offspring, it is not certain that the RGs evaluated to be stably expressed herein remains stable under new experimental conditions. But by closely monitoring RT-qPCR performance, and in particular any changes to mean Ct values for RGs, it is feasible to detect when the RGs are affected by the experimental conditions. In such cases, alternative RGs should be verified to be unaffected by exposure and used instead. By including two to three RGs for the experiments, the chances of correcting for smaller changes to RG regulation is greatly improved contra only using one RG for normalization.

## CONCLUSIONS

In conclusion, we recommend using either of the combinations *Rps18*/*Rpl13a* or *Rps18*/*Ubc* for RT-qPCR experiments on rat cortex following in utero exposure to TH disruptors.

## ACKNOWLEDGEMENTS

The authors like to thank technical staff Heidi Letting, Mette Voigt Jessen, Lillian Sztuk, Stine Marie Stysiek, Ulla El-Baroudy and Dorte Lykkegaard Korsbech for their invaluable contributions, as well as the DTU FOOD animal facilities. The authors also thank the project team: Professor Andreas Kortenkamp, Dr. Martin Scholze and Dr. Olwenn Martin from Brunel University, and the entire Scientific Expert Group affiliated with this project for their very valuable contributions.

### Funding

This work was funded by the European Commission, Directorate-General Environment, Specific contract number 07.0201/2017/769285/ENV.B.2, Implementing Framework contract No ENV.A.3/FRA/2014/0029, and The Danish Environmental Protection Agency. The funders had no role in study design, data collection and analysis, decision to publish, or preparation of the manuscript.

### Grant Disclosures

The following grant information was disclosed by the authors:
European Commission, Directorate-General Environment:
07.0201/2017/769285/ENV.B.2.
Implementing Framework: ENV.A.3/FRA/2014/0029.
The Danish Environmental Protection Agency.

### Competing Interests

The authors declare that they have no competing interests.

### Author Contributions

- Louise Ramhøj conceived and designed the experiments, performed the experiments, analyzed the data, contributed reagents/materials/analysis tools, authored or reviewed drafts of the paper, approved the final draft.
- Marta Axelstad conceived and designed the experiments, performed the experiments, contributed reagents/materials/analysis tools, authored or reviewed drafts of the paper, approved the final draft.
- Terje Svingen conceived and designed the experiments, performed the experiments, analyzed the data, contributed reagents/materials/analysis tools, prepared figures and/or tables, authored or reviewed drafts of the paper, approved the final draft.
## Animal Ethics

The following information was supplied relating to ethical approvals (i.e., approving body and any reference numbers):

The animal study was approved by the Danish Animal Experiments Inspectorate (authorization No. 2015-15-0201-00553 C3) and overseen by the National Food Institute's in-house Animal Welfare Committee for animal care and use.

## Data Availability

The raw data is available in Dataset S1. Data sheets for sample setup, amplification data and results are provided for four groups exposed to thyroid disrupting compounds.

## Supplemental Information

Supplemental information for this article can be found online at http://dx.doi.org/10.7717/peerj.7181#supplemental-information.

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
