# Peer review of "Validation of endogenous reference genes in rat cerebral cortex for RT-qPCR analyses in developmental toxicity studies"

_PeerJ, doi:10.7717/peerj.7181_

## Round 0.1 · original submission · Major Revisions

Please note carefully the comments about things missing. Also, the comment about testing a third gene is especially important, and should be considered. Likewise, addressing assumptions about qPCR, as suggested by Reviewer 1, is important.

·

Basic reporting

English is clear with only small editorial suggestions
1. Line 145, add space after P/N 4303859)…
2. Line 177, add and after this problem…
3. Line 195, add the after upregulation in…
4. Line 203, add space after Ct method…


Tables and figures
1. Statistical significance denotations in the figure legends are missing.
2. In figure 1 the Rps18 and Rpl13a as well as RPS18 and Ubc alone (not in combination) is not plotted.
3. The numbers in figure 1 do not match with the numbers in table 3.

Experimental design

Methods
1. There is no information on how many animals were used other than in table 2. Add to material and methods and figure legends.

Validity of the findings

1. The testing of only two genes for the proof-of-principle gives limited evidence. I would recommend using at least 3 genes of each different expression pattern after exposure (up-regulated, down-regulated, and unchanged) to conclude that more than one reference gene should be used. This could also provide some guidance for the minimum fold-change to consider significant in this experimental setting and add value to the manuscript.
2. I would like to see these genes of interest after all the exposures; PTU, OMC and DE-71 as they might very well show different results and my not support the conclusion statement that these reference genes are to recommend following in utero exposure to TH disruptors.
3. As the authors state, several assumptions are made when performing RT-qPCR (Nolan et al, 2006) and at least some of these should be addressed in this manuscript. At least the RNA quality should be provided and a serial dilution of the samples to assess PCR efficiency and presence of inhibitors.
4. I would also suggest to compare different methods than 2-ΔΔCT for quantification (e.g. Rao et al, 2014 and Pfaffl 2001) as this would add additional value to the manuscript.

Additional comments

This manuscript describes the evaluation of 10 different endogenous reference genes in rat brain after exposure to three putative TH disrupting chemicals. All selected genes were considered stable when analyzed by three different algorithms. The expression of two genes of interest (likely dysregulated after exposure) was assessed using the different reference genes. Pvalb was significantly downregulated regardless of which reference gene was used while Dio2 was only significantly upregulated when three out of the ten reference genes were used for normalization.
The manuscript has in general scientific and methodological soundness. However, it provides no new information and conclusion from the limited amount of genes of interest makes the evidence for the overall conclusion weak. In addition, several quality assurance methods should be added to understand the reliability of the results. Therefore, I would recommend major revision before considered for publication.

Reviewer 2 ·

Basic reporting

Ramhøj et al, have studied validation of reference genes for appropriate normalization of qRT-PCR. The study has potentially interesting data, but the manuscript requires improvement in organization and clarity.

-Abstract is poorly written. Its more like a summary of results. No effort has been taken to underline the problem, in-addition authors have not addressed why 10 genes mentioned were chosen.
-Similar studies have been done in same context they have not been cited.

Experimental design

- Technical samples in triplicates is standard not duplicates as in this study.
- Since only duplicates were used statistical values could be misleading.
-I would like, that authors should clarify the technical and biological replicates.
-The raw data in excel sheet, beta actin is spelled wrong.

Validity of the findings

- Box plots should be used for all samples in figure 2, it give a better understanding of the distribution of the values.
- A plot comparing all 10 RG’s to each other should be included.

---

## Round 0.2 · Minor Revisions

Dear Authors, regarding this comment and. your reply:
Q12: Similar studies have been done in same context they have not been cited.
A12: We are unsure what the Reviewer refers to here. Same context as in exposure scenario, or with regard to cortex tissue? Wherever we have missed valuable information, we would appreciate more details.

I think the reviewer was simply asking for citation of similar studies of reference gene stability in the context of chemical exposures. Please reply as to whether such studies may support the conclusions regarding RGs you have studied.

This may be addressed by a simply reply, with or without additional citations.

Pending that, your manuscript is acceptable.

Thank you for your patience.

---

## Round 0.3 · accepted · Accept

The studies cited do address the concern.

#